# Timeliness Harvesting Loss of Rice in Cold Region under Different Mechanical Harvesting Methods

Jinwu Wang, Xiaobo Sun, Yanan Xu, Wenqi Zhou, Han Tang and Qi Wang *

College of Engineering, Northeast Agricultural University, Harbin 150030, China;
jinwuw@neau.edu.cn (J.W.); sunxiaobo@neau.edu.cn (X.S.); YananXu@neau.edu.cn (Y.X.);
zwq@neau.edu.cn (W.Z.); tanghan@neau.edu.cn (H.T.)
* Correspondence: wangqi@neau.edu.cn

**Abstract:** The yield loss during the process of harvesting is a great challenge in rice production. A suitable harvesting time and harvesting method can help to reduce the yield losses of rice, and decisions about the harvest date have important implications for labor management as well as for agricultural machinery scheduling. Nonetheless, the comprehensive composition of timeliness harvesting loss (THL) and its changing rules for different harvesting methods remain poorly understood. The objective of this study was to determine the effect of harvest date and mechanical harvesting methods on grain dry matter timeliness loss (GDMTL) and mechanical timeliness losses (MTL) of rice in the cold region. To this end, the field experiment was conducted from 45 days after heading (45 DAH) to 59 days after heading (59 DAH), adopting a full-feeding and semi-feeding combine harvester (FCH and SCH) from 2019 to 2020. The results showed that harvest date had a significant effect on GDMTL and four kinds of MTL including header timeliness loss (HTL), cleaning timeliness loss (CTL), un-threshed timeliness loss (UTTL), and entrainment timeliness loss (ETL, only under FCH). With the prolonged harvest date, the HTL and CTL increased and the UTTL and ETL decreased, which ranged from 0.15–0.31%, 0.36–0.67%, 0.72–0.18%, and 0.69–0.31%, respectively for FCH. For SCH, the variation range of HTL, CTL, and UTTL was 0.41–0.59%, 0.66–0.98%, and 0.64–0.21%, respectively. The GDMTL increased first and then decreased, ranging from 2.84–0.87%. The mechanical harvesting methods had no significant effect on the GDMTL of rice, but the MTL could be large between FCH and SCH. In general, optimal harvest period was 52 DAH~53 DAH for both harvesting methods, which exhibited the highest yield and the lowest loss, i.e., 9269.3 kg/hm² and 1.70%, respectively, and the mechanical operating mode on different harvest dates was recommended to minimize the mechanical loss. The optimal harvest date for rice in a cold region ensured both quality and quantity for mechanized harvesting, and provided a reference for the reasonable allocation of operating harvesters in the harvesting season.

**Keywords:** rice; grain dry matter timeliness loss; mechanical timeliness losses; harvesting methods; optimal harvest date



## 1. Introduction

As one of the most important crops, rice serves as the staple food for nearly half of the world's population. Besides, the demand for rice is constantly increasing and, because of this fact, it is important to use proper technology that not only increases production on the same land area, but also preserves biodiversity and the environment [1]. China, as one of the native rice production countries in the world, with an average annual rice planting area and total production of 30.1 million ha and 21.2 billion t, respectively. The total production in China ranked first and the planting area ranked second in the world [2]. In order to ensure the safety of rice production, it is necessary to minimize paddy rice losses in every process of rice production with limited land resources. Timeliness loss in rice production plays an important role as delayed or advanced harvests both cause significant reduction in



the recoverable quantity and quality of rice [3]. Once the harvesting is delayed or advanced, it will inevitably lead to a decline in rice production and constrain resource utilization and environmental sustainability [4,5]. Timely harvesting of rice ensures both good quality and higher yield [6]; however, farmers usually harvest the transplant rice before or beyond the full maturity stage, and the farmers' income is bound to be diminished as a result [7]. Due to the large planting area of rice, mechanized harvesting is essential. However, it is still difficult for rice to be carried out within the suitable harvest period due to the influence of weather, manpower, and harvester dispatching [8]. It is, therefore, essential to evaluate the performance of THL during the harvesting time through appropriate methods to receive maximum yield and quality, which is beneficial to farmers' income and sustainable agriculture development.

The yield loss caused by untimely operation of harvesting is called THL, which performs as a collection of multiple losses including GDMTL and MTL. Many researchers have evaluated the loss of yield during the harvest time; however, no analysis has focused on the factors leading to yield loss and its changing rules. Yoshida [9] concluded that the maximum grain weights were found between 13 and 33 days after heading in most rice cultivars. Surek [10] found that the early harvesting of paddy causes both quantitative and qualitive losses, and the suitable harvesting time was 49 DAH in Northwest Turkey. Vishnu [11] estimated a high quantity of loss of grains per acre during late harvesting followed by mid harvesting and early harvesting, and the loss was found to be the highest at 1.92% for mid-harvest and lowest at 1.74% for early harvest. However, all these studies did not explain the factors affecting rice yield and the variation of the paddy loss, nor did they consider the losses caused by the harvesting machine. Losses due to the combine harvester is one of the main points of interest in considering wastes and losses control [12], and mechanical losses are inevitable during the working process of a combine harvester and can only be reduced in an appropriate way [13]. Some other researchers have assessed the mechanical loss during the process of rice harvesting. However, most of the works reported were about the influence of operating parameters on the amount of mechanical loss, or optimized the mechanical structure of key components [14–17]. No published information is available about the effect of harvest date on the MTL of rice cultivar. Khir [7] assessed the harvesting losses induced by different types of harvesters and the headers under various weather conditions, but did not consider the variation of other mechanical losses. Adisa [18] found that the performance of rice harvesting was highly affected by setting of its critical operating parameters, and gave the optimum setting of rotor height, stripper rotor speed, and forward speed as 270 mm, 17.55 m/s, and 3 km/h, respectively. However, it did not consider that the variation of crop physiological state (moisture content, lodging, etc.) during the harvesting time could lead the optimum operating parameters to change from time to time. Some studies have also discussed the difference in harvesting loss between combine harvesting and manually harvesting [19,20]. With the extensive use of harvesting machine nowadays, it is of little significance to compare with the loss of manually harvesting, but a suitable combine harvester that performs the best during the period of harvesting is worth discussing [21]. However, there is little research on the loss difference under different mechanical harvesting modes. Meanwhile, there is no relevant research to analyze the farmers' economic loss caused by the THL.

Consequently, the objective of this study was to determine the effect of harvest date and mechanical harvesting methods on the GDMTL and MTL of rice in cold regions, and considered the economic performance of rice harvesting. Therefore, a study was undertaken at Acheng Experimental Farm of Northeast Agricultural University to ascertain the optimum stage of harvesting for the transplanting of rice cultivar LG as a factor of the minimum grain loss and the maximum yield under full-feeding combine harvester and semi-feeding combine harvester. The variations in GDMTL, HTL, CTL, ETL, and UTTL during the whole harvesting period from 2019 to 2020 were analyzed, the total loss and the rice yield from 45 DAH to 59 DAH were given to determine the optimum harvest date. In addition, the economic loss caused by the THL was evaluated during the

whole harvest season. The resulting information may serve not only as a guide for the selection of proper mechanical harvesting methods, but also the most desirable period for rice harvesting, and even for the arrangement of the post-harvesting machines such as the straw-returning device, which provides the literature with significant information for farmers and environmental sustainability.

## 2. Materials and Methods

### 2.1. Study Site

The experiment was performed in the Acheng Experimental Farm (127.024 N, 45.529 E) of Northeast Agricultural University in Harbin, Heilongjiang Province in 2019 and 2020 (Figure 1). On the whole, the operation period was 15 days, and was initiated from 45 DAH to 59 DAH. The test area is characterized by a continental monsoon climate in the cold temperate zone. In winter, the climate is cold and dry; in summer, the precipitation is concentrated, and the climate is mild and humid. There are obvious seasonal variations: in spring, the weather warms up quickly, it is mostly windy, and spring drought is easy to cause; summer is short and hot, with a lot of rain, the temperature drops sharply in autumn, and early frost is apt to appear. The annual sunshine hours in the experimental area are 24,421 h on average; the annual average temperature reaches 3.4 °C; the frost-free period is nearly 162 days; the annual average rainfall is 553.2 mm. The cultivated soil in the experimental area has moderate nitrogen content, and an abundant amount of phosphorus, sulfur, iron, manganese, and boron content, whereas potassium and zinc content are lacking. The first crop in the experimental area was rice, which was turned over in autumn. The basic soil fertility in the experimental area is listed in Table 1. The weather and rainfall during the harvest period of the test year are illustrated in Figure 2.

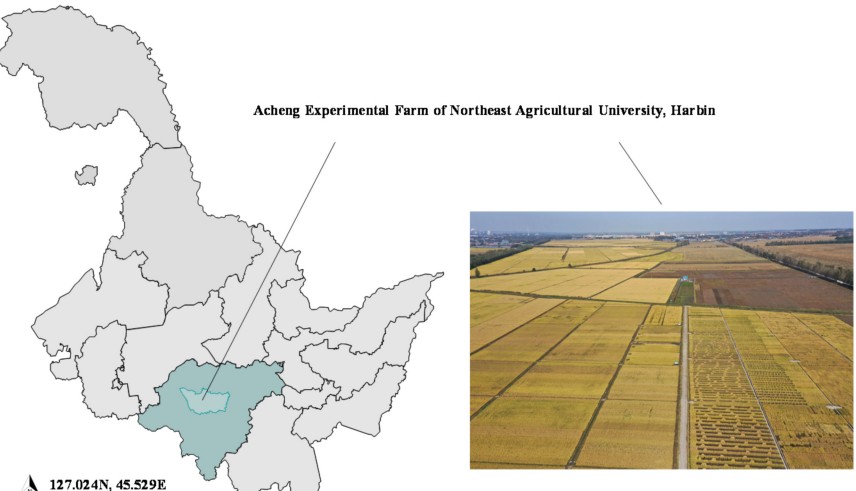

**Figure 1.** Schematic map of the study area.

**Table 1.** The average organic matter content and the soil fertility in the tillage layer.

| Type of the Soil | PH Value | Organic Matter (%) | Alkaline Nitrogen (mg/kg) | Available Phosphorus (mg/kg) | Available Potassium (mg/kg) |
|---|---|---|---|---|---|
| Black paddy soil | 6.74 | 5.21 | 110.8 | 16.1 | 82.1 |

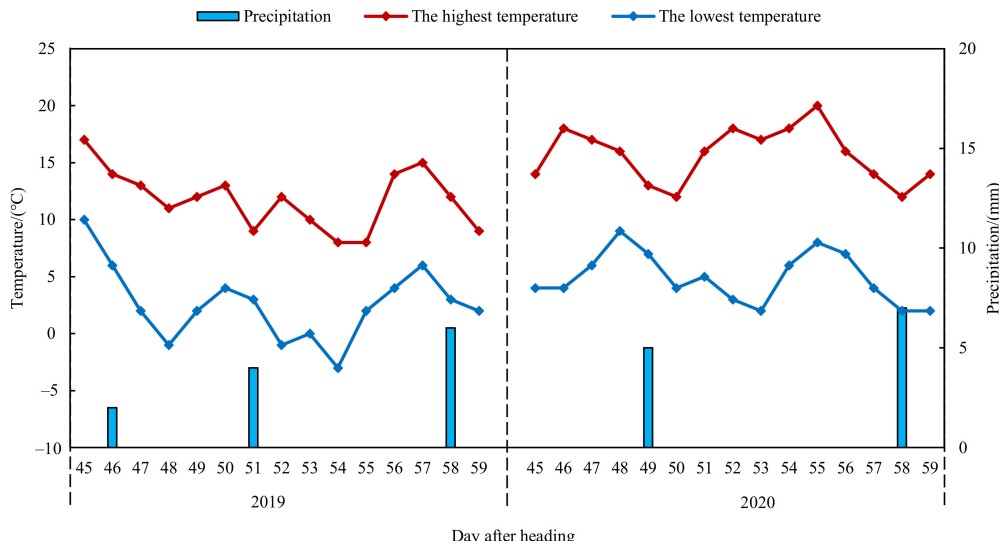

**Figure 2.** The temperature and precipitation during the harvest period in 2019 and 2020.

### 2.2. Test Sample and Field Management

LG, a typical grain rice variety in the cold region of northeast China, was selected as the test variety. Its growth period is about 128 days, and the average spike length is 180 mm. The average grain length is 5.6 mm, the grain width reaches 3.1 mm, and the aspect ratio is at 1.8. It is more suitable for planting in the lower limit of the second accumulated temperate zone and the upper limit of the third accumulated temperate zone in Heilongjiang Province, China. Table 2 lists the basic information of the samples. The seeds were sown on the 10th of April, and seedlings were planted on the 18th of May. The rice transplanter was Kubota 2ZGQ-6D, and the size of rice transplanting was 300 mm × 140 mm. The field management was uniformly performed in compliance with the Standard Rules for high yield and rice production in China Northern Cold Regions [22].

**Table 2.** The properties of the sample.

| Cultivars | Number of Plants/hm$^2$ | Number of Spikes per Plant | Number of Grains per Spike | Seed Setting Rate (%) | Thousand Grain Weight (g) |
|---|---|---|---|---|---|
| LG | $2.56 \times 10^5$ | 14 | 91 | 93 | 28.01 |

### 2.3. Harvesting Instruments and the Parameters Setting

The operating machine for this experiment were the CF805N full-feed rice combine harvester produced by Changfa Agricultural Equipment and the 4LBZ-150 half-feed rice combine harvester from Taihu, Jiangsu. Both machines act as the major methods of local rice harvesting, with their major technical performance and technical parameters listed in Table 3. The Japanese Kate PM-8188-A grain moisture meter and Shanghai Jinghong DHG-9030A electric thermostatic air-blast drying oven were the instruments used to measure the moisture content of rice grains and straw. The Lucky JA203 series electronic balance with a measurement accuracy of 0.01 g was adopted as the instrument to determine the thousand grain weight and the dry and wet mass of the straw. Furthermore, nylon mesh bag (210 cm × 70 cm, 105 cm × 70 cm), rice grain sampling bag (13.5 cm × 6.5 cm × 32 cm), stopwatch (accuracy 0.01 s), 20 m measuring tape (accuracy 0.01 m), thatch cloth (8 m × 8 m), and grain sieves with different meshes were also applied.

**Table 3.** Model and parameters of the combine harvester in the experiment.

| Combine Harvesting Method | Model | Header Width (mm) | Minimum Ground Clearance (mm) | Theoretical Operation Speed (km/h) | Head Cutter Form |
|---|---|---|---|---|---|
| SCH | 4LBZ-150 | 1400 | 100 | 0–5.4 | Standard type II |
| FCH | 4LZ-4 | 2000 | 120 | 0–6.0 | Standard type II |

To test the effect of the harvest date on the mechanical loss of different combine harvesters, the FCH and SCH were ensured to operate under a suitable and constant working conditions. Prior to the test, the header height and engine speed of the FCH and SCH were determined. By complying with the operating procedures of FCH and SCH and the relevant requirements of national standards [23], the engine speed was recommended to be maintained between 2000–3000 r/min, and the stubble height was not higher than 180 mm. To determine the operating conditions of the harvester during the harvest period, the pre-tests were performed to illustrate the total loss rate of the machine at different header heights and engine speeds. Lastly, it can be determined that the optimal operating parameters by using the full-feeding combine harvesting method were the height of the header at 120 mm and an engine speed of 2300 r/min; the optimal operating parameters for using the semi-feeding combine harvest method were the height of the header at 120 mm and an engine speed of 2600 r/min.

*2.4. Experimental Design and Sampling Processes*

For the experiment, the two harvesting methods were set (i.e., FCH and SCH), and the corresponding field test procedures and guidelines were formulated by complying with the national standard [24]. Completely randomized block trials were performed on the samples from 45 DAH to 59 DAH. On the first day of the experiment, most of the grains in the experiment area turned yellow, and some of the stalks were still green. Thus far, all local farmers had not initiated harvesting operations. At the end of the experiment, the rice spikes drooped in the whole field, and the harvesting had been completed by all the local farmers. The test date was extended in accordance with the local natural harvest date. To avoid the boundary effect of the test field, all the rice in the range of 8 m between the test area and the ridge was harvested in advance, and it was convenient for the machine to carry out the driving operation at the same time. Before the operation of the experiment, an area of 20 m² was pre-harvested to preheat the combine harvester. The combine harvesting methods of full-feeding and semi-feeding were repeated five times on the respective harvest date from 45 DAH to 59 DAH. The experimental field was divided into 180 plots and numbered. The length of each plot was over 6 m to minimize the variation and inconsistency of the grain yield and losses [25], and the plot width was greater than the width of the header to ensure full-width harvesting. Finally, each plot was set as 10 m long and 3 m wide. The MATLAB program was adopted to randomly determine each test plot corresponding to the respective harvest date.

The straw crushing device at the rear of the CF805N full-feeding combine harvester was dismantled to collect the rice grains entrained in the effluent and not cleaned. The bolts were used to fix two nylon mesh bags on the rice stalk discharge outlet and the rear of the cleaning sieve to facilitate the collection and selection of the UTTL, ETL, and CTL. The 4LBZ-150 semi-feeding combine harvester was hung with a collection device at the tail to collect the CTL and the UTTL. Before each harvesting experiment, the two combine harvesters were cleared and inspected, and all the sundries in the collection device were emptied. The two combine harvesters were fixed in the standard operating gear during the harvesting. The height of the full-feeding combine harvester was regulated to 120 mm, and the engine speed was fixed at 2300 r/min. The height of the semi-feeding combine harvester was regulated to 120 mm, and the engine speed was fixed at 2600 r/min. After the end of each

strip, the forward power of the machine was cut off, whereas the threshing and cleaning system was maintained until all the rice had completed the threshing and cleaning process. The testing indexes included grain dry matter timeliness loss rate (GDMTLR), header timeliness loss rate (HTLR), entrainment timeliness loss rate (ETLR), cleaning timeliness loss rate (CTLR), and un-threshed timeliness loss rate (UTTLR). The GDMTLR indicates the variation of grain mass, which is an important index that reflects the timeliness harvest loss. The GDMTLR was calculated as Equation (1):

$$\text{GDMTLR} = \frac{DM_{\max} - DM_i}{DM_{\max}} \times 100\% \qquad (1)$$

where $DM_{\max}$ is the maximum dry basis 1000-grain weight during the whole harvesting period (g), and $DM_i$ is the dry basis 1000-grain weight in each plot (g).

As the working process of two combine harvester could be more complex, in the process of combine harvesting, the rice was gathered, conveyed, uploaded, and separated; the rice had already gone through the operation of cutting, threshing, and cleaning. Therefore, the HTL, CTL, ETL, and UTTL were inevitable under these working conditions. The flow chart of the mechanical harvesting process and four kinds of mechanical timeliness loss are shown in Figure 3. The HTL was the drop kernels attributed to the collision of the header during the operation, and the HTLR was calculated by collecting the drop kernels in the non-rutting area of each trip. The ETL referred to the whole grain that was mixing and discharging with the straw, and it was determined by the whole rice kernels that mixed up with the threshed mixture from the back of the threshing cylinder. Since there were only the head of the paddy threshed in the threshing cylinder, there were no ETL during the process of semi-combine harvesting. The CTL indicated the whole grain blown out by the cleaning system, and it was determined by the whole rice kernels that blew out through the cleaning sieves. The UTTL referred to the whole grains being discharged outside the machine without being detached from the ears after the threshing in the threshing cylinder. The ETLR, CTLR, and UTTLR were calculated by collecting the whole grains in those two nylon mesh bags on the rice stalk discharge outlet and the rear of the cleaning sieve.

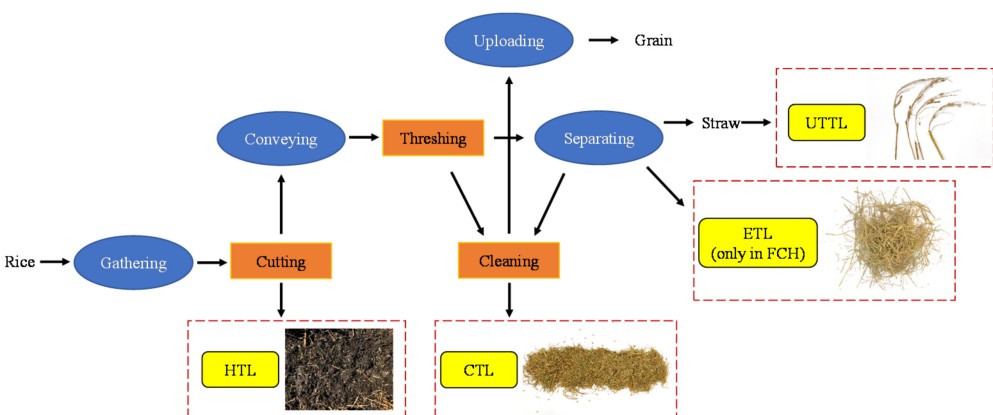

**Figure 3.** The flow chart of mechanical harvesting process.

### 2.5. Statistical Analysis

Statistical analysis was performed using the SPSS 23 statistical package (Statistical Product and Service Solutions Inc., Chicago, IL, USA). ANOVAs were applied to detect significant differences in the loss data (GDMTLR, HTLR, ETLR, CTLR, and UTTLR) during the two harvesting years (2019–2020). Mean comparisons were made by least significant differences (LSD) at the 0.05 probability level. Data were processed and drawn using Microsoft Office Excel.

# 3. Results

## 3.1. Variation of the GDMTLR Based on Different Mechanical Harvesting Methods

The variation of 1000-kernel dry grain weight with the harvest date by applying different combine harvesting methods is presented in Figure 4. As indicated by the multiple comparison analysis of LSD, the 1000-kernel dry grain weight increased first and then decreased under the two harvesting methods. The 1000-kernel dry grain weight was the lowest in the period of 45 DAH–46 DAH, and reached the maximum in the period of 51 DAH–54 DAH. At the late-harvest period, the 1000-kernel dry grain weight decreased significantly from 57 DAH to 59 DAH. During the whole harvesting period, no significant difference was reported in the 1000-kernel dry grain weight on any two consecutive days. Given the variations in the1000-kernel dry grain weight in the whole experiment period (45 DAH–59 DAH), the whole harvest period could be divided into five stages, i.e., ST1 (45 DAH–46 DAH), ST2 (47 DAH–50 DAH), ST3 (51 DAH–54 DAH), ST4 (55 DAH–56 DAH), and ST5 (57 DAH–59 DAH) (Figure 4). The average 1000-kernel dry grain weight at the five stages from ST1 to ST5 reached 24.04 g, 24.42 g, 24.75 g, 24.53 g, and 24.33 g, respectively. As indicated by Figure 4, during the harvest period of the experiment year (2019–2020), ST1~ST3 were the accumulation stages of grain dry matter. During this period, the 1000-kernel dry grain weight increased significantly by 2.93%. ST3~ST5 were the consumption stages of grain dry matter; during this period, the 1000-kernel dry grain weight declined significantly by 1.69%. As revealed above, the GDMLR was the largest in ST1, and the average loss rate was 2.84%. The GDMLR in ST2 declined, with an average loss rate of 1.31%. The GDMLR in ST3 was the lowest, for the 1000-kernel dry grain weight reached its' peak at this stage. With the extension of the harvest date, the 1000-kernel dry grain weight in ST4 and ST5 tended to increase, and the average GDMLR reached 0.87% and 1.69%, respectively. No significant difference was reported in the 1000-kernel dry grain weight by applying different combine harvesting methods on the identical day. Thus, the combine harvesting method had no significant effect on the variation of grain dry matter.

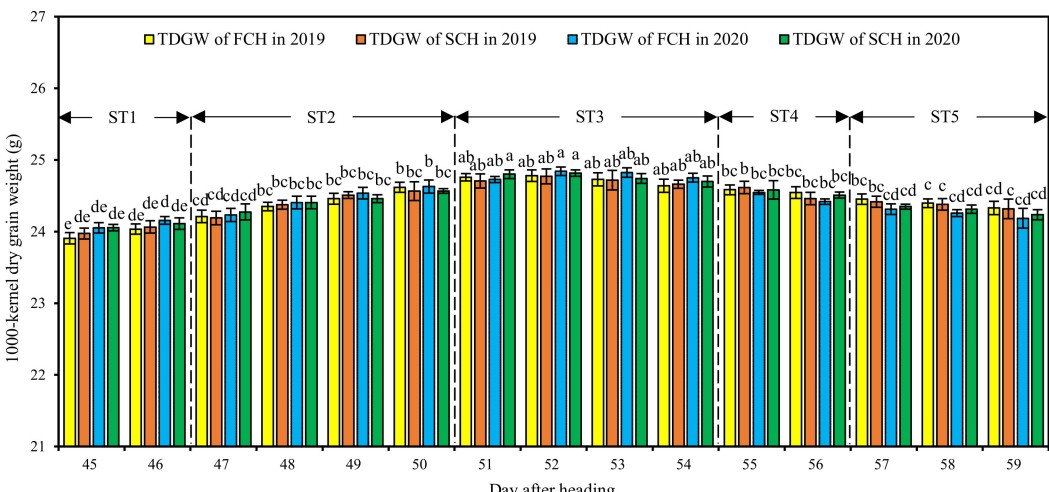

**Figure 4.** The variation of 1000-kernel dry grain weight with harvest date under FCH and SCH in 2019 and 2020. Note: Different lowercase letters in the figure indicate significant differences at the level of *p* = 0.05 between treatments, the same as below.

## 3.2. Variation of the MTL Based on Different Mechanical Harvesting Methods

Figure 5 presents the variations of HTLR, CTLR, ELR, and UTLR of LG during the harvest period (45 DAH–59 DAH) of the experimental year (2019–2020) by using FCH and SCH. Since only the spikes of the paddy entered the threshing cylinder under semi-combine harvesting, and the straw was outside of the harvesting machine without contact with grains, no ETL was identified under the SCH, and the ETL was only analyzed when employing the full-feeding combine harvester. The results showed that there was no

significant difference in HTLR and CTLR based on the two combine harvesting methods from 45 DAH to 49 DAH; among them, the average values of HTLR and CTLR by employing FCH were 0.15% and 0.27%, respectively; the average values of HTLR and CTLR by applying SCH reached 0.41% and 0.58%, respectively, which were all at the lowest level during the entire harvest period. With the extension of the harvest date, the HTLR and CTLR of the two harvesting methods increased significantly from 50 DAH to 59 DAH, except for the rainy days. Among them, the variation range of HTLR by applying FCH ranged from 0.15% to 0.31%, that of CTLR ranged from 0.36% and 0.67%, and the average daily growth rate reached 8.65% and 7.27%, respectively. The variation range of HTLR under SCH was 0.41–0.59%, for CTLR it was 0.66–0.98%, and the average daily growth rate was 3.81% and 3.97%, respectively. The HTLR and CTLR were significantly different between the two combine harvesting methods. Compared with the method of full-feeding combine harvesting, the HTLR and CTLR of the semi-feeding combine harvester were 0.31% and 0.35% higher, respectively. Rainfall during the harvesting period would result in a sharp decline in HTLR and CTLR; among them, the HTLR by employing FCH decreased to 0.12–0.23%, and the CLR decreased to 0.22–0.36%. The HTLR under SCH decreased to 0.34–0.53%, while the CLR decreased to 0.54–0.81%. The harvest date had a significant effect on UTLR and ELR under FCH, and both losses tended to decrease except for on the rainy days. Moreover, the range of UTTLR under FCH was from 0.69% to 0.31%, and the average daily decline rate reached 6.27%. The variation range of UTTLR by applying SCH ranged from 0.64% to 0.21%, and the average daily decline rate was 9.09%. The UTLR of the full-feeding combine harvester was 0.17% higher than that of the semi-feeding combine harvester. The variation range of ETLR by using FCH ranged from 0.72% to 0.18%, and the average daily decline rate was 10.34%. Rainfall during the harvest period would cause the UTTLR and the ETLR to rise sharply. Furthermore, on those rainy days, the range of UTTLR under SCH ranged from 0.67% to 0.42%, and the UTTLR and ETLR under FCH rose sharply to 0.77–0.59% and 0.84–0.58%, respectively.

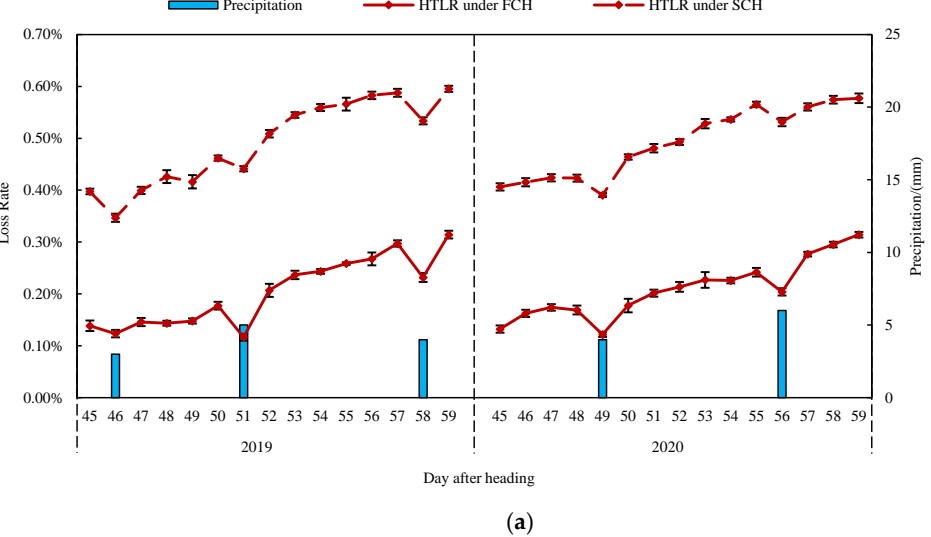

(**a**)

**Figure 5.** *Cont.*

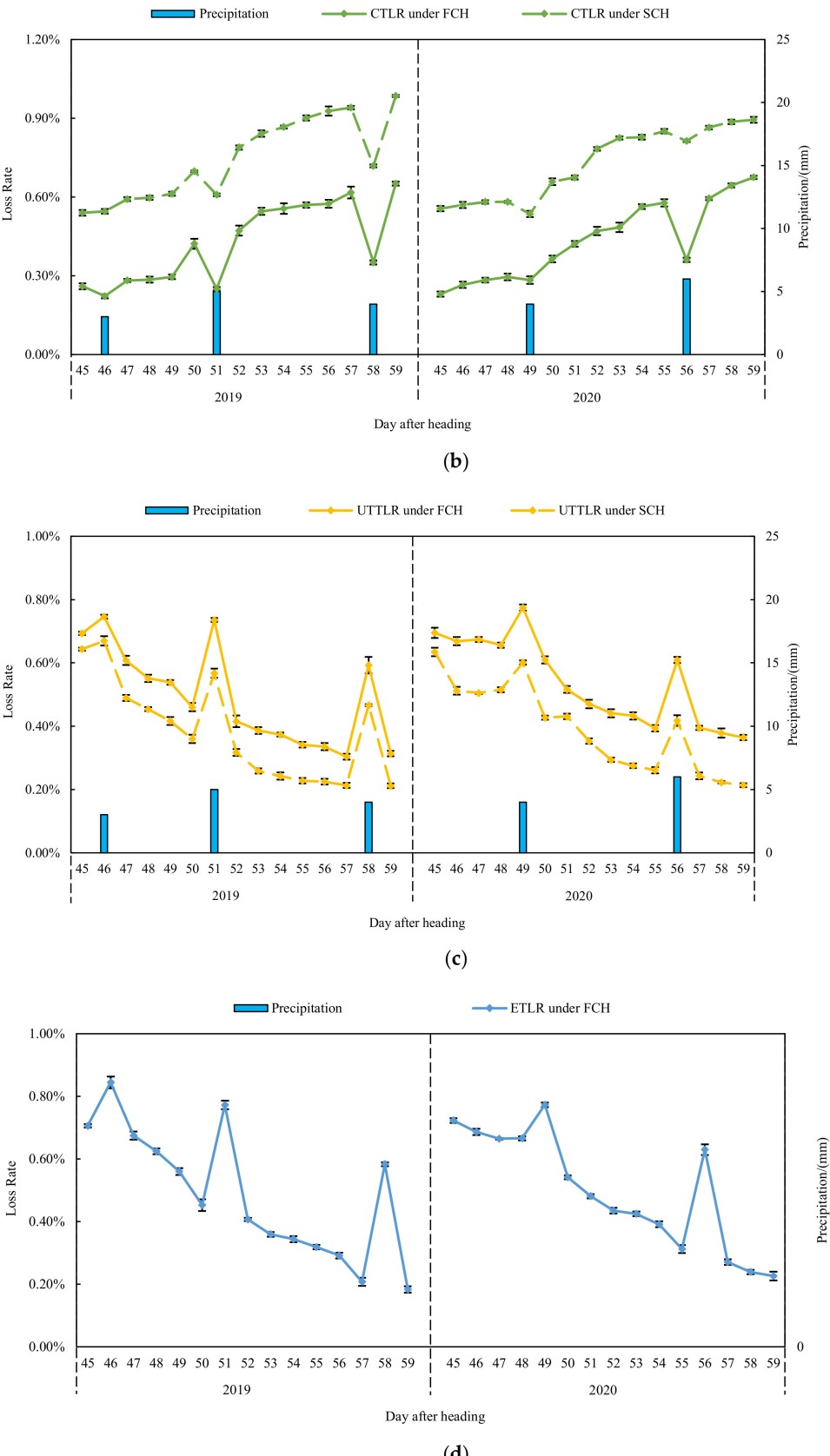

**Figure 5.** The variation of (**a**) HTLR; (**b**) CTLR; (**c**) UTTLR; (**d**) ETLR with harvest date under FCH and SCH in 2019 and 2020.

*3.3. Variation of the Timeliness Harvest Loss and the Differences in the Proportion of Each Loss*

Figure 6 illustrates the variation of the total loss of LG during the two consecutive years from 45 DAH to 59 DAH based on the full-feeding and semi-feeding combine harvesting methods and the differences in the loss ratios. It is therefore suggested that the total timeliness harvest loss first decreased and then increased during the whole experimental period. The total timeliness harvesting loss rate by using the full-feeding combine harvesting method ranged from 5.33% to 1.50%, and the range of mechanical timeliness harvest loss rate was 1.94% to 1.46%. The total timeliness harvesting loss rate of the semi-feeding combine harvester ranged from 4.80% to 1.61%, and the range of mechanical timeliness harvest loss rate ranged from 1.80% to 1.50%. During the harvesting period of the experimental year (2019–2020), the mechanical loss rate of the two combine harvesting methods complied with the requirements of National Standards [23]. In each harvesting day, the total timeliness harvesting loss rate of FCH was 0.21% higher than that of the SCH. Moreover, based on the two harvesting methods, the GDMTL acted as the main source of the total timeliness harvest loss during the pre-harvest period (45 DAH~50 DAH) and the late-harvest period (55 DAH–59 DAH), which took up 67.08–26.97%. The loss of GDMTL could only be reduced by reasonably regulating the harvesting date. Thus, the harvesting operation that was carried out at the mid-harvest period (51 DAH–54 DAH) could avoid the large losses, and the MTL was the main issue at this period. During this period, the proportions of various mechanical losses under different harvesting methods varied greatly. By employing the full-feeding combine harvesting method, ETL and CTL acted as the main source of total losses at the mid-harvest period, which accounted for 23.00–16.52% and 12.71–28.43%, respectively. For the semi-feeding combine harvesting method, the CTL was the main source of total losses at the mid-harvest period, which took up nearly 32.39–48.85%. Accordingly, reducing the cleaning loss to obtain a higher yield was also the main goal of the semi-feeding combine harvesting method from 51 DAH to 54 DAH. In addition, the proportion of each loss in the total losses varied with the extension of the harvest period. As shown in Table 4, the comparison of each loss rate in different harvest periods can be seen clearly. For the method of full-feeding combine harvesting, each loss rate during the pre-harvest period (45 DAH–50 DAH) was GDMTLR > ETLR > UTTLR > CTLR > HTLR; and it turned to CTLR > UTTLR > ETLR > HTLR > GDMTLR at the mid-harvest period (51 DAH~54 DAH); and finally turned to GDMTLR > CTLR > UTTLR > ETLR > HTLR at the late-harvest period (55 DAH–59 DAH). Based on the semi-feeding combine harvesting method, each loss rate in the pre-harvest period (45 DAH–50 DAH) was GDMTLR > CTLR > UTTLR > HTLR; and it turned to CTLR > HTLR > UTTLR > GDMTLR at the mid-harvest period (51 DAH–54 DAH); and finally turned to GDMTLR > CTLR > HTLR > UTTLR at the late-harvest period (55 DAH–59 DAH). The operating time and operating parameters of the operating machinery should be regulated reasonably in accordance with the variations in different harvesting methods in different harvesting periods to obtain the optimal harvest yield.

**Table 4.** Comparison of each loss rate in different harvest period under FCH and SCH.

| Harvesting Method | Harvest Period | | |
|---|---|---|---|
| | **45 DAH to 50 DAH** | **51 DAH to 54 DAH** | **55 DAH to 59 DAH** |
| FCH | GDMTLR > ETLR > UTTLR > CTLR > HTLR | CTLR > UTTLR > ETLR > HTLR > GDMTLR | GDMTLR > CTLR > UTTLR > ETLR > HTLR |
| SCH | GDMTLR > CTLR > UTTLR > HTLR | CTLR > HTLR > UTTLR > GDMTLR | GDMTLR > CTLR > HTLR > UTTLR |

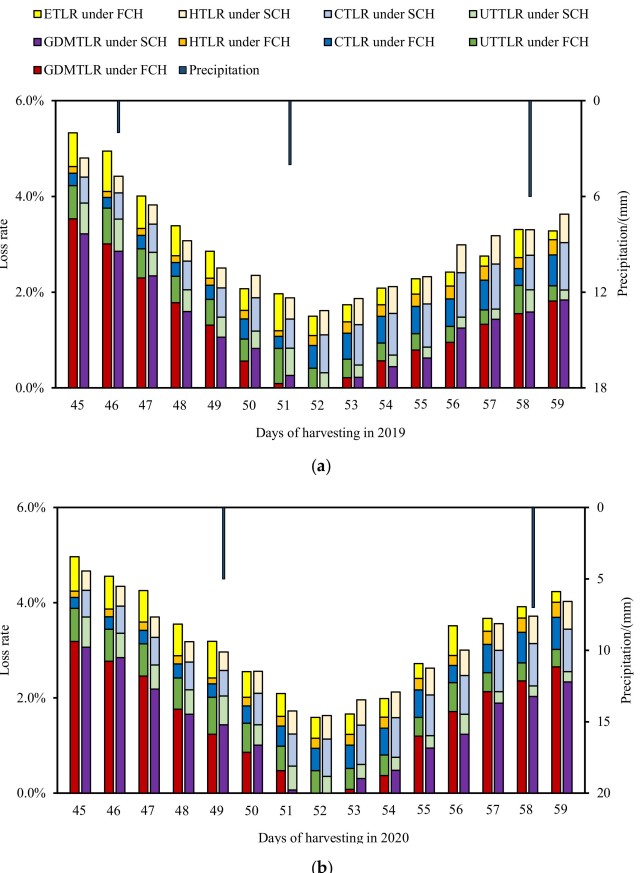

**Figure 6.** The variation of total loss rate and proportion difference of GDMTLR, HTLR, CTLR, ETLR and UTTLR under FCH and SCH in (**a**) the year of 2019 and (**b**) the year of 2020.

### 3.4. The Effect of Harvest Date on Yield under Different Combine Harvesting Methods

Based on the full-feeding and semi-feeding combine harvesting methods, Table 5 lists the GDMTLR, MTL, and the mechanical harvesting yield (MHY) of LG during the harvesting period (45 DAH–59 DAH) from 2019 to 2020. As indicated from this table, the mechanical harvesting method had no significant effect on the yield of rice in cold regions, and year of harvesting showed no significant effect on both GDMLR and MTL. However, the harvest date had a significant effect on rice yield; moreover, with the extension of the harvest date, the rice yield first increased and then decreased. As can be seen from Table 5, the MHY increased constantly from 45 DAH to 52 DAH and ranged from 8904.5 kg/hm$^2$ to 9303.1 kg/hm$^2$; however, the MHY showed a trend of decrease from 52 DAH to 59 DAH and ranged from 9303.1 kg/hm$^2$ to 9018.1 kg/hm$^2$. The variation of the MHY was mainly due to the changing of GDMLR during the whole harvesting period, for the GDMLR was the largest on 45 DAH which was 3.25% on average, and it decreased to the minimum on 52 DAH; when it came to the last day of harvesting the GDMLR appeared to be increased to 2.16% on average. The fluctuation of MTL was little during the whole harvesting period, which was 1.64% on average. Overall, advanced or delayed harvesting would result in great yield loss. Furthermore, the rice harvesting from 45 DAH to 46 DAH achieved the lowest rice yield, varying from 8904.5 kg/hm$^2$ to 9010.0 kg/hm$^2$, with an average total loss rate of 3.68%. The rice yield of 47 DAH~49 DAH and 57 DAH–59 DAH ranked the second, ranging from 9011.5 kg/hm$^2$ to 9191.0 kg/hm$^2$, and the average total loss rate reached 2.44%. The rice yield of 50 DAH~51 DAH and 54 DAH–56 DAH increased, ranging from 9045.3.5 kg/hm$^2$ to 9256.1 kg/hm$^2$, which achieved an average total loss rate of 1.21%. During the whole harvesting period, 52 DAH–53 DAH achieved the highest rice yield and the lowest total loss rate under FCH and SCH. Thus, it is therefore determined that the optimal harvest date of LG is 52 DAH–53 DAH.

**Table 5.** The variation of GDMLR, MTL, and MHY with harvest date under FCH and SCH in 2019 and 2020.

| DAH | Year of Harvesting | Harvesting Method | GDMLR /(%) | MTL /(%) | MHY /(kg/hm$^2$) | DAH | Year of Harvesting | Harvesting Method | GDMLR /(%) | MTL /(%) | MHY/ (kg/hm$^2$) |
|---|---|---|---|---|---|---|---|---|---|---|---|
| 45 | 2019 | FCH | 3.53 | 1.80 | 8904.5 ± 15.0 e | 53 | 2019 | FCH | 0.21 | 1.53 | 9232.8 ± 19.0 ab |
| | | SCH | 3.22 | 1.58 | 8982.9 ± 48.8 de | | | SCH | 0.22 | 1.65 | 9258.8 ± 21.7 ab |
| | 2020 | FCH | 3.19 | 1.78 | 8942.6 ± 20.3 e | | 2020 | FCH | 0.08 | 1.58 | 9264.4 ± 23.8 ab |
| | | SCH | 3.07 | 1.60 | 8988.4 ± 52.2 de | | | SCH | 0.31 | 1.65 | 9245.1 ± 27.3 ab |
| 46 | 2019 | FCH | 3.01 | 1.94 | 8920.8 ± 37.7 e | 54 | 2019 | FCH | 0.57 | 1.52 | 9211.1 ± 25.5 bc |
| | | SCH | 2.86 | 1.56 | 8956.5 ± 27.3 e | | | SCH | 0.44 | 1.67 | 9253.1 ± 32.3 ab |
| | 2020 | FCH | 2.77 | 1.78 | 8980.2 ± 17.4 de | | 2020 | FCH | 0.37 | 1.62 | 9221.5 ± 13.8 b |
| | | SCH | 2.85 | 1.50 | 9010.0 ± 21.3 de | | | SCH | 0.48 | 1.64 | 9224.7 ± 23.6 b |
| 47 | 2019 | FCH | 2.30 | 1.71 | 9022.5 ± 14.7 de | 55 | 2019 | FCH | 0.79 | 1.49 | 9181.8 ± 16.2 bc |
| | | SCH | 2.34 | 1.48 | 9070.5 ± 11.3 cd | | | SCH | 0.63 | 1.70 | 9234.6 ± 20.4 ab |
| | 2020 | FCH | 2.46 | 1.80 | 9011.5 ± 25.8 de | | 2020 | FCH | 1.20 | 1.53 | 9166.2 ± 21.8 bc |
| | | SCH | 2.19 | 1.51 | 9066.1 ± 19.1 cd | | | SCH | 0.95 | 1.68 | 9187.1 ± 21.2 bc |
| 48 | 2019 | FCH | 1.78 | 1.60 | 9079.5 ± 24.0 cd | 56 | 2019 | FCH | 0.95 | 1.47 | 9160.6 ± 13.7 bc |
| | | SCH | 1.60 | 1.48 | 9135.5 ± 18.4 c | | | SCH | 1.25 | 1.74 | 9175.4 ± 44.3 bc |
| | 2020 | FCH | 1.76 | 1.79 | 9080.8 ± 21.5 cd | | 2020 | FCH | 1.71 | 1.80 | 9045.3 ± 39.8 d |
| | | SCH | 1.66 | 1.52 | 9109.0 ± 26.6 cd | | | SCH | 1.24 | 1.67 | 9100.5 ± 24.2 cd |
| 49 | 2019 | FCH | 1.31 | 1.54 | 9136.6 ± 20.8 c | 57 | 2019 | FCH | 1.33 | 1.43 | 9118.3 ± 21.4 cd |
| | | SCH | 1.06 | 1.45 | 9191.0 ± 23.5 bc | | | SCH | 1.43 | 1.74 | 9153.9 ± 28.1 bc |
| | 2020 | FCH | 1.24 | 1.95 | 9080.0 ± 23.6 cd | | 2020 | FCH | 2.13 | 1.54 | 9062.3 ± 25.9 cd |
| | | SCH | 1.44 | 1.53 | 9091.7 ± 13.2 cd | | | SCH | 1.89 | 1.67 | 9097.0 ± 22.7 cd |
| 50 | 2019 | FCH | 0.56 | 1.51 | 9191.8 ± 32.3 b | 58 | 2019 | FCH | 1.55 | 1.76 | 9061.9 ± 23.6 cd |
| | | SCH | 0.83 | 1.52 | 9221.9 ± 30.0 b | | | SCH | 1.58 | 1.72 | 9086.0 ± 24.6 cd |
| | 2020 | FCH | 0.86 | 1.69 | 9182.2 ± 27.2 bc | | 2020 | FCH | 2.36 | 1.69 | 9043.7 ± 22.6 d |
| | | SCH | 1.01 | 1.55 | 9193.8 ± 21.4 bc | | | SCH | 2.03 | 1.99 | 9076.5 ± 32.7 cd |
| 51 | 2019 | FCH | 0.09 | 1.87 | 9192.5 ± 25.0 bc | 59 | 2019 | FCH | 1.82 | 1.46 | 9084.3 ± 19.8 cd |
| | | SCH | 0.26 | 1.62 | 9195.8 ± 28.0 bc | | | SCH | 1.84 | 1.80 | 9108.9 ± 43.6 cd |
| | 2020 | FCH | 0.47 | 1.62 | 9215.2 ± 20.1 b | | 2020 | FCH | 2.65 | 1.58 | 9018.1 ± 30.1 de |
| | | SCH | 0.07 | 1.66 | 9256.1 ± 30.6 ab | | | SCH | 2.34 | 1.69 | 9051.4 ± 30.1 d |
| 52 | 2019 | FCH | 0.00 | 1.50 | 9275.5 ± 11.7 ab | | | | | | |
| | | SCH | 0.00 | 1.62 | 9303.1 ± 60.4 a | | | | | | |
| | 2020 | FCH | 0.00 | 1.59 | 9283.1 ± 14.0 ab | | | | | | |
| | | SCH | 0.00 | 1.63 | 9291.4 ± 23.6 ab | | | | | | |

Note: Different lowercase letters in the figure indicate significant differences at the level of $p$ = 0.05 between treatments.

## 4. Discussion

The 1000-kernel dry grain weight under FCH and SCH both increased first and then decreased with the extending of the harvest date. This difference was not significant in two adjacent harvesting days, however, significant variations occurred among different stages (ST1~ST5) during the harvest period. This was mainly because the accumulation and consumption of dry matter in rice grains was due to the conversion of stem and leaf assimilation and the consumption of self-respiration [26,27]. This variation process was generally slow, and the cool weather in Northeast China could also delay the growth of crops [28]. Accordingly, many related studies have recorded variations in grain weight at intervals of 5 to 10 days during the harvest period. Kwon et al. [29] conducted an experiment every 5 days during the harvest period and obtained that the maximum rice yield occurred on the day of 30 DAH, which was the best date for rice harvest. Hossein et al. [30] set two levels of harvest time on the 29th of August and the 8th of September, and determined the optimal harvest day as the 29th of August through the rice plant height and yield obtained on the two harvest days. However, a large test interval could result in a large gap between the optimal harvest period and the actual optimal harvest date, and the test results were not accurate. Thus, this study conducted daily harvest tests during the harvest period to ensure a complete observation of the dynamic variations of various losses during the harvest period. During the harvest period of the experimental year, the 1000-kernel dry grain weight from ST1 to ST3 increased gradually. This was mainly because rice leaves still maintained a high leaf area index in the early stage of harvest, and photosynthesis was still strong [31]. The dry basis thousand-grain weight of rice grains was closely related to the net photosynthetic capacity of leaves. Moreover, over two-thirds of the dry matter accumulation of rice grains originated from photosynthesis after flowering [32–34]. Accordingly, the rice grains in the pre-harvest stage remained in the growing stage, and early harvesting would cause greater losses due to insufficient dry matter accumulation in the rice grains. The dry basis thousand-grain weight of rice grains from ST3 to ST5 tended to decrease. This was primarily due to the gradual senescence of crop leaves at the later stage of harvest, meaning that the photosynthetic area of the leaves that could be used to support growth decreased, and photosynthesis was weakened [35,36]. Moreover, respiration, carbohydrate transport, and starch synthesis of crops had certain energy requirements. As a result, the dry weight of rice grains at the late harvest stage decreased. During the whole harvest period, the 1000-kernel dry grain weight of rice grains peaked at ST3. Then, the accumulation of leaves due to photosynthesis and the consumption of grains due to respiration was balanced, and the rice grains reached their fullest state.

In terms of the HTLR, the HTLR under full-feeding and semi-feeding combine harvesting methods tended to increase with the extension of the harvest date. Since the whole field rice spikes appeared to sag gradually with the maturity of rice, the collision and friction between the spikes and the cutting platform could be more intense [37]. Thus, the falling grains increased under the external force, and the HTLR increased significantly [38]. On the other hand, with the extension of the harvest date, the environmental temperature dropped sharply, and it was always accompanied by relatively large north winds during the harvest season. These factors could cause the SMC decline. Figure 7 illustrates the variations in the SMC of the LG during the experimental year (2019–2020). As indicated by Figure 7, SMC not only gradually decreased with the extension of the harvest date, but also decreased within each harvesting day. The decrease of SMC could cause the connection force between the grain and the stalk to decrease gradually. When the operating parameters of the harvester remained unchanged, the rice grains could fall off more easily [39] and the HTLR increased gradually. A significant difference was identified in HTLR between FCH and SCH. This difference was mainly due to the different arrangement of headers for the two harvesting methods. When the full-feeding combine harvester was operating, the crop moved backwards to the cutter through the reel. The whole crop was cut and transported to the feeding inlet through the auger. During the transportation, most of the grains were falling into the header frame, so the HTLR was small under FCH. The cutting table of the

semi-feeding combine harvester adopted a series of reeling fingers and chains to realize the orderly clamping and conveying of the crop. During this period, the entire straw was exposed to the outside of the body, and the clamping process could cause a large shattering loss, which will eventually lead to a large amount of HTLR.

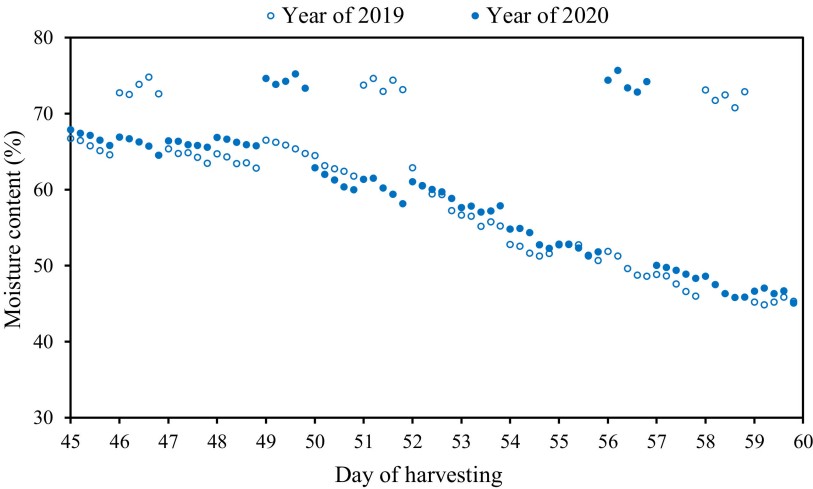

**Figure 7.** The variation of SMC with harvest date in 2019 and 2020.

The CTLR based on full-feeding and semi-feeding combine harvesting methods tended to increase with the extension of the harvest date. The amount of the CTLR was mainly related to the wind speed of the fan and the opening of the cleaning screen. Excessive wind speed or an excessively large cleaning sieve angle would destroy the balance between the friction and gravity components of the grain, and eventually cause the missing of intact rice grains [40,41].

Moreover, with the extension of the harvest date, the decrease of GMC would also cause the wet-basis grain weight to decrease. Accordingly, at the identical fan speed and opening degree of the cleaning screen, the grains could be more easily blown out by the fan. Moreover, the decrease of moisture content could lead to the decrease of adhesion with the sieve plate [42,43], thereby causing the decrease in the friction between the grain and the sieve plate, and increasing the CTLR. Rainfall during the harvest period could result in a significant reduction in the CTLR of the two harvesting methods. This was primarily explained by the fact that rainfall could increase the grain mass and the friction between the sieve and grains would increase for the increase of moisture content, which could make it more difficult for intact grains to be blown off the sieve. Furthermore, the layout of the cleaning devices of the two harvesters were identical. However, the full-feeding combine harvester fed the whole crop into the threshing cylinder and the semi-feeding combine harvester only threshed the spikes. As a result, the load on the cleaning sieve during the full-feeding combine harvesting was greater, and the intact grains were more difficult for the fan to blow out. Thus, the CTLR of SCH was larger than that of FCH. This was also the main reason that the harvest date had a more significant impact on the CTLR of the full-feeding combine harvesting method.

In terms of the UTTLR loss, the UTTLR based on full-feeding and semi-feeding combine harvesting methods showed a trend of decline with the extension of the harvest date. In the initial period of harvesting, the SMC were relatively large, resulting in a relatively humid environment in the threshing room, and the threshing performance declined. Accordingly, the ETLR at the initial stage of harvesting was relatively large. However, with the extension of the harvest date, the moisture content of the head of rice decreased gradually, which led to a decrease in spike activity and tensile strength; the grains on the spikes fell off more easily [44,45], which eventually caused a gradual decrease in UTTLR. For the method of full-feeding combine harvesting, the head and the straw were impacting and rubbing together in the threshing cylinder, and the higher threshing

load and the SMC would cause the threshing performance to decrease. For the method of semi-feeding combine harvesting, only the head of rice entered the threshing cylinder. The threshing load was relatively small, however the threshing performance still maintained a high level. Thus, the UTTLR under the FCH was larger than the SCH.

In terms of ETLR, the ETLR in adopting the full-feeding combine harvesting method showed a trend of decrease with the extension of the harvest date. The main reason was that the higher moisture content at the early stage of harvesting could enhance the adhesion between the grains and the straw [46,47]. As a result, rice grains were more likely to be entrained and discharged out of the rear of the machine by the straw, so the ETLR at the early stage of harvesting was larger. With the prolonged harvest date, the SMC decreased, and the adhesion between the grains and the straw decreased, and eventually the ETLR decreased as well. Since the straw did not enter the threshing cylinder under the method of semi-feeding combine harvesting, there was no ETLR under SCH. This was also the main reason why the loss rate of the FCH during 45 DAH–49 DAH was higher than that of the SCH. At the final stage of harvesting, the ETLR under FCH decreased sharply, and the MTL of the SCH was gradually larger than that of the FCH. During the whole harvesting period, advanced harvesting or rainfall could increase the moisture content in the threshing room, which would eventually increase the ETLR. Accordingly, the ETLR at the early stage of harvesting was relatively large and the rainfall could cause the ETLR to increase sharply.

In brief, the total loss rate of LG during the pre-harvest period (45 DAH–50 DAH) and the late-harvest period (55 DAH–59 DAH) by employing SCH and FCH were higher, which was primarily attributed to the larger GDMLR. The lowest total loss rate was observed during the mid-harvest period (51 DAH–54 DAH), which largely resulted from the lowest GDMLR at this stage. In addition, the economy performance was considered during the whole harvesting period: the average price of the test variety was 2.6 yuan/kg in the year of 2019 and 2020 [48], and the economic loss caused by timeliness harvesting could be serious. Adopting the harvesting method of FCH, the economic loss of the rice harvesting from 45 DAH to 46 DAH, 47 DAH to 49 DAH, 50 DAH to 51 DAH, 54 DAH to 56 DAH, and 57 DAH to 59 DAH, was 850.0 yuan/hm$^2$, 508.2 yuan/hm$^2$, 178.2 yuan/hm$^2$, 258.8 yuan/hm$^2$, and 517.9 yuan/hm$^2$, respectively. However, for the method of SCH, the economic loss was 754.4 yuan/hm$^2$, 426.3 yuan/hm$^2$, 150.0 yuan/hm$^2$, 204.6 yuan/hm$^2$, and 465.4 yuan/hm$^2$, respectively. The economic loss of FCH was larger than that of SCH for its bigger MTL at each stage of harvesting. Thus, to reduce the rice yield loss and the economic loss during the whole harvesting period, an appropriate harvest time for operation was essential. However, due to the limited number of the harvesting machine, most farmers compete for farming hours and the harvesting operations were performed at the wrong time. At this time, the MTL should be minimized. For instance, the operating speed should be reduced to decrease the feeding amount, as an attempt to minimize the ETLR under FCH [49,50]. Moreover, the air volume of the cleaning fan and the angle of the cleaning screen should be reduced appropriately, and the rotating speed of the threshing cylinder should be increased to improve the quality of grain separation, as an attempt to reduce the CTLR and UTTLR under FCH and SCH [51–53], which help to attain the largest yield during the harvest period. This study could provide references for time management during the process of rice harvesting, as well as guidance for rice straw management after mechanical harvesting. The next step of our research would focus on the post-harvesting methods and its economic performance, such as straw-returning methods. The study of pre-harvesting and post-harvesting could be beneficial to environmental sustainability.

## 5. Conclusions

According to the characteristics of the various losses of rice based on different mechanical harvesting methods and the harvest date, this paper discussed the comprehensive composition of timeliness harvesting loss (THL) and its changing rules under FCH and SCH. The conclusions are as follows:

The harvest date significantly impacted the GDMTL. With the extension of the harvest date, the 1000-kernel dry grain weight first increased and then decreased, and it peaked in ST3 (51 DAH–54 DAH). The GDMLR first decreased and then increased. The variation in the GDMTL is correlated to the accumulation and consumption of dry matter in rice grains, and it is mainly affected by the date of harvesting.

For the MTL during the whole harvesting period (45 DAH–59 DAH) from 2019 to 2020, the HTL and CTL increased, and the UTTL and ETL decreased. A significant difference could be identified in the MTL under different mechanical harvesting methods. On each harvesting day, the HLR and CLR of the SCH were higher than that of the FCH, however the UTLR was lower. The variation in the MTL is correlated to the decline of the SMC during the whole harvesting season.

The total loss rate under the two mechanical harvesting methods first decreased and then increased during the whole harvesting period. The GDMLR was the main source of the total harvest loss during the pre-harvest period (45 DAH–50 DAH) and late-harvest period (55 DAH–59 DAH). The total loss rate in the middle of harvest (51 DAH–54 DAH) was the smallest, and MTL was the main source of total loss. Besides, the mechanical harvesting method insignificantly impacted rice yield. Moreover, the two machine harvesting methods both achieved the maximum output in 52 DAH–53 DAH, which is the optimal harvest date during the whole season and the total loss rate was the smallest at the same time. Accordingly, a suitable harvesting date is the most important factor that affect the rice yield, and a reasonable deployment of operating machinery should focus on when the harvesting operation is advanced or delayed. Moreover, the economy loss of FCH was larger than that of SCH for its bigger MTL at each stage of harvesting. Consequently, the harvesting method of SCH is more strongly recommended for rice harvesting in cold regions.

**Author Contributions:** Conceptualization, J.W. and Q.W.; methodology, X.S.; software, X.S.; validation, W.Z., and H.T.; formal analysis, Q.W. and Y.X.; investigation, J.W.; resources, J.W.; data curation, X.S. and Y.X.; writing—original draft preparation, X.S.; writing—review and editing, Q.W.; visualization, X.S.; supervision, Q.W.; project administration, J.W.; funding acquisition, J.W. All authors have read and agreed to the published version of the manuscript.

**Funding:** This work was funded by National Natural Science Foundation of China (32071910); Construction Project of National Industrial Technology System (CARS-01-44); Heilongjiang Collaborative Innovation Promotion System for Rice Production (HCIPC-RP).

**Institutional Review Board Statement:** Not applicable.

**Informed Consent Statement:** Not applicable.

**Data Availability Statement:** The authors confirm that the data supporting the findings of this study are available within the article, and the data in detail are available on request from the corresponding author, for its restrictions on policy.

**Acknowledgments:** Measurements in from the harvesting season 2019 to 2020 were done with the support from the team of Intelligent Paddy Field Equipement. We offer the heartfelt thanks to their help in performing the field experiment.

**Conflicts of Interest:** We declare that we have no known competing financial interest or personal relationships that could have appeared to influence the work reported in this paper.

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
