# Peer review of "Timeliness Harvesting Loss of Rice in Cold Region under Different Mechanical Harvesting Methods"

_sustainability, doi:10.3390/su13116345_

Round 1
Reviewer 1 Report
I found the topic of the study very interesting and in line with the scope of the journal. To improve the overall quality of the manuscript, I have some suggestion/comments as below:
The quality of the figures 1, 2, 4, 5 and 6 may be improved, at least in my pdf they are getting a bit distorted.
Need a better explanation in Table 2. The variation of GDMLR, MTL, and MHY with harvest date under FCH and SCH in 2019 and 2020, It is hard to understand it and you should comment on the values showed in the table.
Table 2. The variation of GDMLR, MTL, and MHY with harvest date under FCH and SCH in 2019 and 2020, instead of Table 2, it should be replaced by Table 4. The variation of GDMLR, MTL, and MHY with harvest date under FCH and SCH in 2019 and 2020.
Figure 4 The flow chart of mechanical harvesting process, need a better explanation in the text.
Section 3.3. Variation of the timeliness harvest loss and the differences in the proportion of each loss, in lines 342-351 are not clear and needs to be better explained in the text.
English needs to be revised.
Author Response
As the content of the reply is more detailed, please see the attachment, thankyou.

Reviewer 2 Report
The manuscript titled “Timeliness Harvesting Loss of Rice in Cold Region under Different Mechanical Harvesting Methods” by Jinwu Wang et, al. provides significant detail of the role of timeline and harvesting method in rice yield and losses.
Here are few comments and suggestions:
The ‘Methods’ and the ‘Results’ section are not written properly. Many of the explanation are mentioned in the result sections. The authors have started the result section with figure 5, which is somewhat unusual. Please explain the result section from figure 1. In figure 3, author should explain the significance of the two figures or if not necessary, should be removed.
Figure legends are also not written properly. Explain the figure in its legend.
Figure 7: explain the color code and alphabets A and B in the figure.
The phrase ‘to be specific’ has been extensively used throughout the manuscript. In my opinion, all the information should be specific and precise and need not to be mentioned.
The author could consider to add a table of comparative loss rate in different harvest period.
In Discussion: author used many future tenses in the sentences, which is confusing. Author may need to rewrite the discussion.
Author Response
As the content of the reply is more detailed, please see the attachment, thankyou!
